# Linearization of Composite Material Damage Model Results and Its Impact on the Subsequent Stress–Strain Analysis

**DOI:** 10.3390/polym14061123

**Published:** 2022-03-11

**Authors:** Jarmil Vlach, Radek Doubrava, Roman Růžek, Jan Raška, Jan Horňas, Martin Kadlec

**Affiliations:** VZLU—Czech Aerospace Research Center, 199 05 Prague, Czech Republic; vlach@vzlu.cz (J.V.); doubrava@vzlu.cz (R.D.); ruzek@vzlu.cz (R.R.); raska@vzlu.cz (J.R.); hornas@vzlu.cz (J.H.)

**Keywords:** composite, Hashin, linearization, damage simulation, linear solver, non-linear solver

## Abstract

To solve problems in the field of mechanical engineering efficiently, individual numerical procedures must be developed, and solvers must be adapted. This study applies the results of a carbon-fibre reinforced polymer (CFRP) analysis along with the nonlinear finite element damage (FE) method to the translation of a linear solver. The analyzed tensile test sample is modelled using the ply-by-ply method. To describe the nonlinear post-damage behavior of the material, the Hashin model is used. To validate the transformation, an analysis and comparison of the damage results of the linearized and nonlinear model is carried out. Job linearization was performed by collecting elements into groups based on their level of damage and pairing them with unique material cards. Potentially suitable mathematical functions are tested for the grouping and consolidation of the elements. The results show that the agreement of some presented methods depends on the damage level. The influence of the selected statistical functions on the result is shown here. The optimal solution is demonstrated, and the most efficient method of linearization is presented. The main motivation behind this work is that the problem has not been discussed in the literature and that there is currently no commercial software translator that provides the transference of models between solvers.

## 1. Introduction

In the field of mechanical engineering, there are several problems that need to be solved by numerical methods. To increase their efficiency, individual procedures have been developed for individual tasks and solvers have been adapted. The most common variants of the solution include a linear or nonlinear variant of the calculation following an explicit or implicit scheme [1,2]. Individual calculation schemes differ in mathematical formulation, stability and speed of computation. Linear solvers are based on the assumption that the force in the model increases with increasing deformation during the calculation [1,2]. If this condition is violated, the task becomes poorly conditioned and the calculation cannot be successfully performed [3]. However, the phenomenon of force drop in the model occurs in all tasks that are focused on the study of the state of structures in a broken state [4,5,6,7] beyond the strength reached before total structural collapse [8,9,10,11,12]. This can occur, for example, due to a collision between fast-moving objects [13]. For these reasons, violation tasks are calculated with the help of solvers following the so-called explicit scheme. They do not require the condition of increasing forces in the model; they are stable but only suitable for simulations of fast or quasi-static processes. Because the tasks for individual solvers are formulated differently, it is necessary to ensure appropriate continuity between them. Thus, the order of calculations, which are based on different calculation schemes, cannot be created completely arbitrarily. A good example is the sequence of analyses to determine the stiffness of a composite part before and after failure [14], which is the focus of this article, because changes in stiffness can be based on a structural condition monitoring (SHM) system that can determine the damage location of a local stiffness change. First, a linear approach following the implicit scheme is used, then a nonlinear analysis with violation is performed using the explicit solver, and then again, a linear analysis with the implicit scheme follows. As an example, in this work, a method of transferring information from the results of the explicit solver to the model of the implicit solver Abaqus 6.14 is presented. The linearization method is used to transfer information between individual models. The main motivation of this study is that there is no communication between solvers and there are not enough resources to address a similar issue in such a direct way, as addressed in this work. Furthermore, a commercial software translator providing the translation of models does not currently exist. This paper focuses mainly on composite material failure, such as has been referred to in previous studies [15,16,17,18,19,20]. In general, the linearization (model conversion) procedure can be applied to all materials and implicit solvers. The aim of the article is to find a unique and highly efficient linearization method, which is theoretically verified.

## 2. Methodology

### 2.1. Workflow Overview

The workflow diagram is shown in Figure 1. The experimental simulation following the standardized ASTM D3039 test was chosen for demonstration [20]. The problem contains a material description at the level of Hooke’s law [1,3] and follows Hashin’s progressive damage model [1,17,19]. The results are subject to translation, which mainly concerns the material description at the level of individual elements of the model. According to the degree of damage, the damaged elements are consolidated into groups with help of mathematical functions and paired with unique material cards, which are calculated for each group. Furthermore, an analysis and comparison of linear and nonlinear models is performed. The success of consolidation functions is compared in a demonstration task.

In principle, the individual formulations differ in one specific way in terms of material description. While the nonlinear violation model contains mathematical functions with violation parameters, in the linear model, nonlinearities are replaced by a larger number of material cards with a linear response. The consolidation functions listed below provide the matching key between the material cards and the violation parameters.

### 2.2. Basic Calculus

In this paper, the components of the stress and strain tensor in the form shown by Equation (1) are considered for each finite element [1]. This corresponds to the formulation for tasks in the field of composite materials, solved with the help of the Abaqus system and with the use of shell-type elements. Components of tensors that are perpendicular to the plane of the element are not considered, so each contains only five components instead of the usual six.
(1)$$\left\{\tilde{\sigma }\right\}=\left\{\begin{array}{c} {\tilde{\sigma }}_{11}\\ {\tilde{\sigma }}_{22}\\ {\tilde{\sigma }}_{12}\\ {\tilde{\sigma }}_{13}\\ {\tilde{\sigma }}_{23} \end{array}\right\},\;\left\{\hat{\sigma }\right\}=\left\{\begin{array}{c} {\hat{\sigma }}_{11}\\ {\hat{\sigma }}_{22}\\ {\hat{\sigma }}_{12}\\ {\hat{\sigma }}_{13}\\ {\hat{\sigma }}_{23} \end{array}\right\},\left\{\sigma \right\}=\left\{\begin{array}{c} \sigma_{11}\\ \sigma_{22}\\ \sigma_{12}\\ \sigma_{13}\\ \sigma_{23} \end{array}\right\},\left\{\varepsilon \right\}=\left\{\begin{array}{c} \varepsilon_{11}\\ \varepsilon_{22}\\ \varepsilon_{12}\\ \varepsilon_{13}\\ \varepsilon_{23} \end{array}\right\}$$

The linear formulation of the stress and strain analysis problem is shown by Equation (2) [1,18,19], where $\left[C^{0}\right]$ in Equation (3) represents the initial stiffness matrix of the composite material without including the possible failure effect.
(2)$$\left\{\tilde{\sigma }\right\}=\left[C^{0}\right].\left\{\varepsilon \right\}$$
(3)$$\left[C^{0}\right]=\left[\begin{array}{ccccc} \frac{E_{1}^{0}}{1-\nu_{12}^{0}\nu_{21}^{0}} & \frac{\nu_{21}E_{1}^{0}}{1-\nu_{12}^{0}\nu_{21}^{0}} & 0 & 0 & 0\\ \frac{\nu_{12}E_{1}^{0}}{1-\nu_{12}^{0}\nu_{21}^{0}} & \frac{E_{2}^{0}}{1-\nu_{12}^{0}\nu_{21}^{0}} & 0 & 0 & 0\\ 0 & 0 & G_{12}^{0} & 0 & 0\\ 0 & 0 & 0 & G_{13}^{0} & 0\\ 0 & 0 & 0 & 0 & G_{23}^{0} \end{array}\right]$$

The nonlinear formulation of the problem shown by the relationship between material damage and effective stress [1] can be calculated using Equations (4) and (5), which shows the damage operator $\left[D\right]$ [1,6,7,10,11,12].
(4)$$\left\{\hat{\sigma }\right\}=\left[D\right].\left\{\tilde{\sigma }\right\}$$
(5)$$\left[D\right]=\left[\begin{array}{ccccc} \frac{1}{\left(1-d_{1}\right)} & 0 & 0 & 0 & 0\\ 0 & \frac{1}{\left(1-d_{2}\right)} & 0 & 0 & 0\\ 0 & 0 & \frac{1}{\left(1-d_{12}\right)} & 0 & 0\\ 0 & 0 & 0 & \frac{1}{\left(1-d_{13}\right)} & 0\\ 0 & 0 & 0 & 0 & \frac{1}{\left(1-d_{23}\right)} \end{array}\right]$$

The stress calculation of the damaged composite layer in simplified form is also performed using the Equations (6) and (7) [1], where $\left[C^{d}\right]$ represents a stiffness matrix with damage rates $d_{1}$, $d_{2}$, $d_{12}$, $d_{13}$ and $d_{23}$, similar to Equation (4); however, this formula does not include the influence of Poisson’s ratio decrement, such as in (7) [9].
(6)$$\left\{\sigma \right\}=\left[C^{d}\right].\left\{\varepsilon \right\}$$
(7)$$\left[C^{d}\right]=\left[\begin{array}{ccccc} \frac{\left(1-d_{1}\right)E_{1}^{0}}{1-\left(1-d_{1}\right)\left(1-d_{2}\right)\nu_{12}^{0}\nu_{21}^{0}} & \frac{\left(1-d_{1}\right)\left(1-d_{2}\right)\nu_{21}^{0}E_{1}^{0}}{1-\left(1-d_{1}\right)\left(1-d_{2}\right)\nu_{12}^{0}\nu_{21}^{0}} & 0 & 0 & 0\\ \frac{\left(1-d_{1}\right)\left(1-d_{2}\right)\nu_{12}^{0}E_{2}^{0}}{1-\left(1-d_{1}\right)\left(1-d_{2}\right)\nu_{12}^{0}\nu_{21}^{0}} & \frac{\left(1-d_{2}\right)E_{2}^{0}}{1-\left(1-d_{1}\right)\left(1-d_{2}\right)\nu_{12}^{0}\nu_{21}^{0}} & 0 & 0 & 0\\ 0 & 0 & \left(1-d_{12}\right)G_{12}^{0} & 0 & 0\\ 0 & 0 & 0 & \left(1-d_{13}\right)G_{13}^{0} & 0\\ 0 & 0 & 0 & 0 & \left(1-d_{23}\right)G_{23}^{0} \end{array}\right]$$

The degree of shear damage can be expressed by the Equation (8) [1], where $d_{1}^{+}$, $d_{1}^{-}$, $d_{2}^{+}$ and $d_{2}^{-}$ represent the rates of damage in tension and pressure.
(8)$$d_{s}=d_{23}=d_{13}=d_{12}=1-\left(1-d_{1}^{+}\right)\left(1-d_{1}^{-}\right)\left(1-d_{2}^{+}\right)\left(1-d_{2}^{-}\right)$$

The damage rates in the main directions of anisotropy $d_{1}$, $d_{2}$ and their relation to the effective stress show the Equations (9) and (10) [1,8].
(9)$$d_{1}=\left\{\begin{array}{c} d_{1}^{+},\;\;\;{\hat{\sigma }}_{11}\geq 0\\ d_{1}^{-},\;\;\;{\hat{\sigma }}_{11}<0 \end{array}\right.$$
(10)$$d_{2}=\left\{\begin{array}{c} d_{2}^{+},\;\;\;{\hat{\sigma }}_{22}\geq 0\\ d_{2}^{-},\;\;\;{\hat{\sigma }}_{22}<0 \end{array}\right.$$

Equation (11) shows that the stresses calculated using the linear and nonlinear formulation are equal up to the ultimate strength of the material.
(11)$$\left\{\tilde{\sigma }\right\}=\left\{\hat{\sigma }\right\},\;\;\;\;\mathrm{if}\;\left\{d_{1};d_{2},d_{12}\right\}=0$$

### 2.3. Graphic Interpretation of Hashin’s Violation

A graphical interpretation of the stress response of the composite material to deformation in the main direction of anisotropy is shown in Figure 2 [1,6]. The diagram does not express all the key points of Hashin’s theory, instead showing only those that are essential in this paper and for the solver used [1].

Before failure, the linear solver follows the blue line without stiffness changes with material stiffness $E_{i}^{0}$, which is the same in tension and pressure. The nonlinear solver proceeds when breaking a composite material according to Hashin’s theory, which is based on the assumption that the breaking can take place independently of the meaning, direction and plane of anisotropy. Once the material strength limit is reached, the material stiffness is gradually lost as a result of the failure. The accumulation of a larger number of failures then leads to a depletion of “damage energy” and a complete loss of rigidity. Thus, reformulating the problem for the linear solver has obvious pitfalls. There may be a situation where a total failure occurs, or the material is damaged in tension ($d_{i}^{+}$) and pressure ($d_{i}^{-}$) in different ways. This is reflected in all elastic constants. These can then be unique for each element or their group, depending on the degree of damage. The linear solver in post-damage analyses uses only one stiffness value and proceeds according to the curve $E_{i}^{d}$. In the presented method, we assume that we can categorize groups of elements into independent groups according to the degree of damage. Unlike the nonlinearity of elastic properties, the uniqueness of material properties is not an obstacle for a linear solver. When transforming a problem with a violation, it is therefore necessary to reformulate the results, especially in terms of material properties.

### 2.4. Consolidation Function

The damage of each element is characterized by four state variables: $d_{1}^{+}$, $d_{1}^{-}$, $d_{2}^{+}$ and $d_{2}^{-}$, which represent its size, direction and meaning. The consolidation function has two roles. First, it must facilitate the association of state variables and the categorization of elements according to the degree of damage. The main goal of this work was to find and verify suitable consolidation functions *F*1, *F*2 and *F*12, with which it is possible to effectively perform parameter reduction. Functions were chosen and sorted with respect to the complexity of their formulation and the truth of the response. The state variables and their consolidation scheme are shown in Table 1.

A total of five functions were tested, which are listed in Table 2. These are basic statistical functions or functions that can be easily derived from them.

The arithmetic mean was calculated using the Equation (12).
(12)$$d_{i,a}=\frac{\left|d_{i}^{+}\right|+\left|d_{i}^{-}\right|}{2},\;\;\;\;\mathrm{for}\;i=\left\{1,\;2\right\}$$

The weighted average was calculated using Equation (13), where $\left\|d_{i}\right\|$ represents the standard calculated according to (14). This is used in the calculation of the weights $w_{i}^{+}$ a $w_{i}^{-}$ in Equation (15).
(13)$$d_{i,w}=\frac{w_{i}^{+}.d_{i}^{+}+w_{i}^{-}.d_{i}^{-}}{w_{i}^{+}+w_{i}^{-}},\;\;\;\;\mathrm{for}\;i=\left\{1,\;2\right\}$$
(14)$$\|d_{i}\|=\sqrt{{\left(d_{i}^{+}\right)}^{2}+{\left(d_{i}^{-}\right)}^{2}},\;\;\;\;\mathrm{for}\;i=\left\{1,\;2\right\}$$
(15)$$w_{i}^{+}=\frac{d_{1}^{+}}{\left\|d_{i}\right\|},\;w_{i}^{-}=\frac{d_{i}^{-}}{\left\|d_{i}\right\|},\;\;\;\;\mathrm{for}\;i=\left\{1,\;2\right\}$$

The consolidation of shear damage is found using the function *F*12, which takes the form of form (8) for all methods.

### 2.5. Material Card Generation

This research shows that perhaps the most accurate way to translate the results of a nonlinear problem into a linear problem is to assign a unique material card to each element. The number of elements in the model then corresponds to the number of material cards. However, this approach places extremely high demands on the preprocessor, making the task confusing.

Therefore, the material cards were generated according to the diagram shown in Figure 3, which represents different variations of damage with a 10% step for each meaning and direction. The variation in damage in one direction can then be represented by an array of 11 rows and 11 columns. However, given the stability conditions (Table 3), a real material card cannot exist for zero 100% damage. At the extreme intervals, in the context of stability conditions [1,18], we must consider a maximum damage value of 99.99%. As the results show, this consideration is sufficient for demonstration purposes. The field then contains exactly 11 × 11 items, which represents a total of 121 material cards.

Assuming the validity of the conditions given for Equations (1)–(11), the elastic properties of the individual material cards can be calculated using (16)–(22), for which the properties are listed in Table 4. Other shear modules can be calculated in an analogous way. The derivation of Equation (16) is given in the literature [18,19], where its validity is considered only within the material without damage. In this contribution, the validity of the relationship is extended to all material cards, including material cards with violations.
(16)$$\left\{\varepsilon \right\}=\left[S^{d}\right].\left\{\sigma \right\}$$
(17)$$S^{d}={\left(C^{d}\right)}^{-1}$$
(18)$$\nu_{12}^{d}=-\frac{S_{21}^{d}}{S_{11}^{d}},\;\;\;\;{\mathrm{if}\;\sigma }_{11}\neq 0,\;\sigma_{22}=0,\;\sigma_{12}=0$$
(19)$$\nu_{21}^{d}=-\frac{S_{12}^{d}}{S_{22}^{d}},\;\;\;\;{\mathrm{if}\;\sigma }_{11}=0,\;\sigma_{22}\neq 0,\;\sigma_{12}=0$$
(20)$$E_{1}^{d}=\frac{1}{S_{11}^{d}},\;\;\;\;{\mathrm{if}\;\sigma }_{11}\neq 0,\;\sigma_{22}=0,\;\sigma_{12}=0$$
(21)$$E_{2}^{d}=\frac{1}{S_{22}^{d}},\;\;\;\;{\mathrm{if}\;\sigma }_{11}=0,\;\sigma_{22}\neq 0,\;\sigma_{12}=0$$
(22)$$G_{12}^{d}=\frac{1}{S_{33}^{d}},\;\;\;\;{\mathrm{if}\;\sigma }_{11}=0,\;\sigma_{22}=0,\;\sigma_{12}\neq 0$$

### 2.6. Material Properties

For test tasks, a material with elastic parameters was used, which are listed in Table 5. Strength parameters respecting the rules of Hashin’s violation in the Abaqus system are shown in Table 6. Parameters describing the progressive development of the violation are given in Table 7 [14].

### 2.7. Damage Variation Matrix and Response of Consolidation Functions

Figure 4, Figure 5, Figure 6 and Figure 7 show the relationship of the consolidation functions to the damage variation according to the diagram shown in Figure 3.

The graphical display illustrates the consequence of the relationship of the selection group of elements at the level of one direction of damage. While Figure 4 presents the consolidation of the damage parameter in the direction of the main diagonal, Figure 5 shows the direction of the side diagonal. Figure 6 describes the variation of damage in the top row direction, and Figure 7 displays the direction of the left column of the variation scheme. It is obvious that the damage values can take values that are only in the interval from 0 to 1.

### 2.8. Testing Examples

The simulation of the laminate sample tensile test was carried out according to ASTM D3039. The sample dimensions of the experiment diagram are shown in Figure 8. At point A, the sample was completely fixed; at point B, a kinematic excitation was introduced, which follows the course shown in Figure 9.

The demonstration task was modelled in all cases with the help of S4R shell elements with a Lamina material formulation and a mesh size of 1 mm. The sample was modelled using the ply-by-ply method. In the case of the explicit solver, the layers were connected by a cohesive contact without failure; in the case of a linear solution, it was a linear contact.

In the first cycle, the sample was loaded in the linear region and the initial stiffness was calculated. The second cycle was designed to break the sample. This part of the task was solved only using Abaqus/Explicit. The third cycle was again simulated using both solvers. Table 8 shows the laminate compositions on which the process was verified.

Although it is only a simulation of the tensile test, due to the contraction of differently oriented undamaged layers of the laminate, local significant pressure damage also occurs. The occurrence of this well-known accompanying phenomenon is checked, because otherwise the validation would only be partially valid. This is also the main reason as to why the method is tested on different layouts.

## 3. Results and Discussion

In Figure 10, the initial stiffness of samples determined by calculation is shown. The result is the same stiffness for calculations obtained from linear and explicit solvers. As shown in Figure 11, the results of the percentage decrease in the stiffness of the tested samples with respect to the reference stiffness.

The agreement of the residual stiffness results, compared with a reference value, is shown in Figure 12. The deviance from the reference solution is referenced in Table 9. It represents a comparison between the initial stiffness calculated with a linear solver and residual stiffness calculated by an explicit solver. It is visible that the decrease in stiffness depends on the composite layup, while the applied load cycle is the same. This was the aim, because the influence of fibre orientation has been considered. In this case, the result of the nonlinear simulation of cycle 1 was considered to be the reference value.

In Figure 13, Figure 14, Figure 15 and Figure 16, the groups of undamaged laminate elements (Layup 4), created based on the use of the presented consolidation functions, are shown.

It is important to note that where there was no (Figure 13) or complete damage, all consolidating functions except the arithmetic mean results were the same. This is consistent with Figure 4, Figure 5, Figure 6 and Figure 7. Thus, it means that Layup 1 and Layup 6 and 7 should theoretically have the same results; however, this did not occur. This was caused by the results of the linearization of several elements which are not completely damaged (though may be partially damaged). This error cumulates and because the stiffness depends on the spatial distribution of the element, the deviation from the true value is very high. This small difference is shown in Figure 14c,d and Figure 15c,d. It is visible that the number of elements is nearly the same, but only the cracks in Figure 14d, Figure 15d and Figure 16d are correct and acceptable.

## 4. Conclusions

It has been shown that in the case of small-scale damage, the majority of the residual stiffness is formed by the undamaged layer; therefore, the agreement of the results is still very good. However, in cases of significant damage, the tested statistical functions fail. In all cases, only the product consolidation function can work properly. It was also found that the reduction in the amount of information caused by the idealization of material cards does not have a significant effect on the result, unlike the choice of the selection function. The pairing of material cards with selected groups of elements, which was carried out in the way described, is unique, highly efficient and accurate. The calculation was stable in all cases.

## Figures and Tables

**Figure 1 polymers-14-01123-f001:**
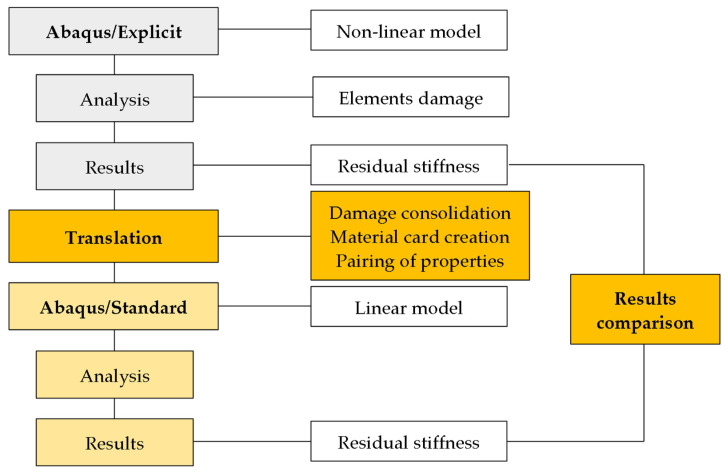
Workflow schema.

**Figure 2 polymers-14-01123-f002:**
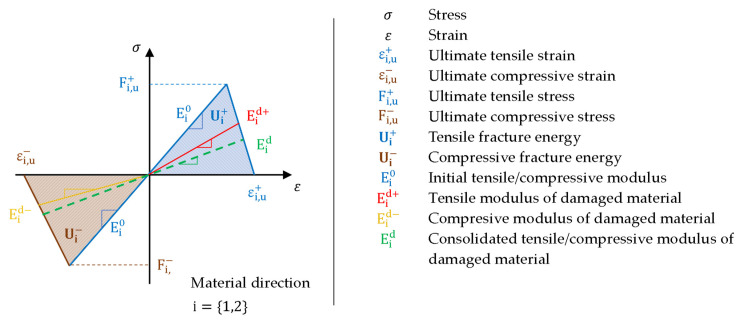
Material stiffness response.

**Figure 3 polymers-14-01123-f003:**
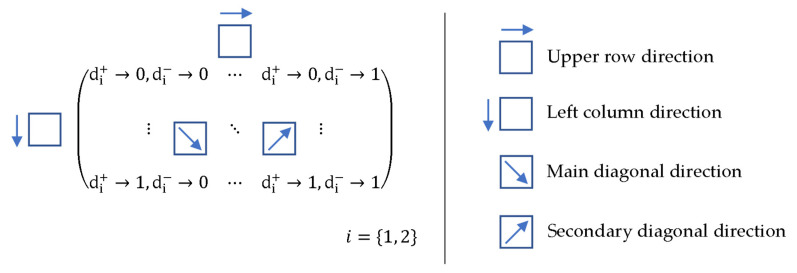
Schema of damage variation.

**Figure 4 polymers-14-01123-f004:**
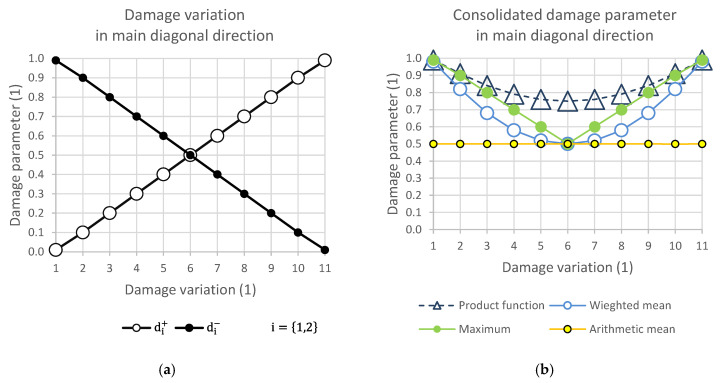
(**a**) Damage variation in main diagonal direction; (**b**) consolidated damage parameter.

**Figure 5 polymers-14-01123-f005:**
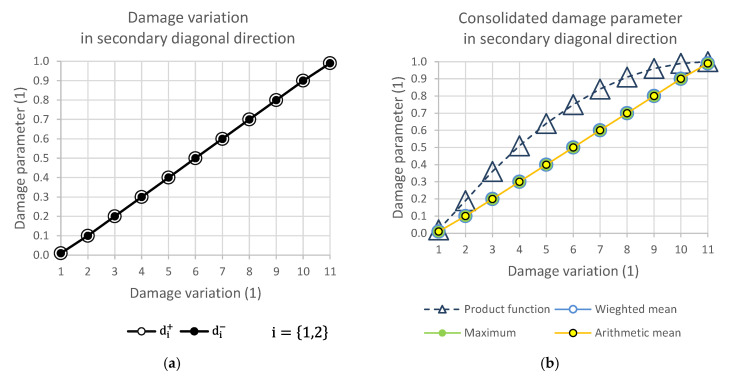
(**a**) Damage variation in secondary diagonal direction; (**b**) consolidated damage parameter.

**Figure 6 polymers-14-01123-f006:**
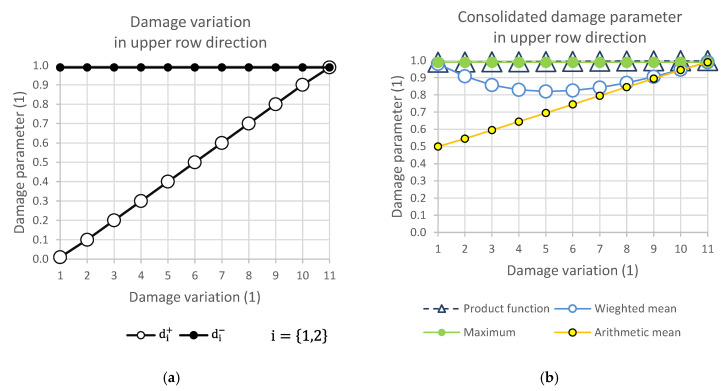
(**a**) Damage variation in upper row direction; (**b**) consolidated damage parameter.

**Figure 7 polymers-14-01123-f007:**
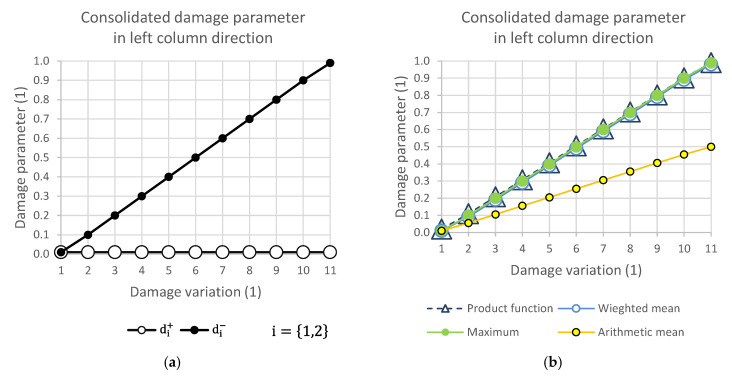
(**a**) Damage variation in left column direction; (**b**) consolidated damage parameter.

**Figure 8 polymers-14-01123-f008:**
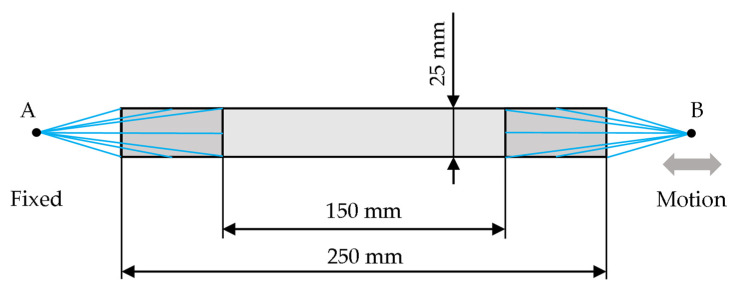
Test coupon, ASTM D3039.

**Figure 9 polymers-14-01123-f009:**
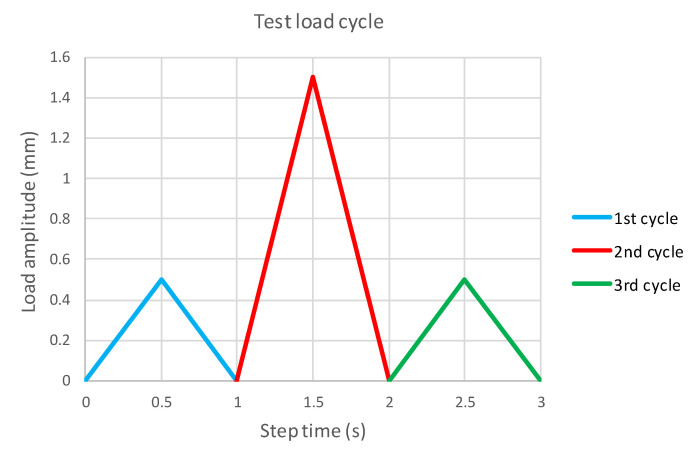
Point B motion history.

**Figure 10 polymers-14-01123-f010:**
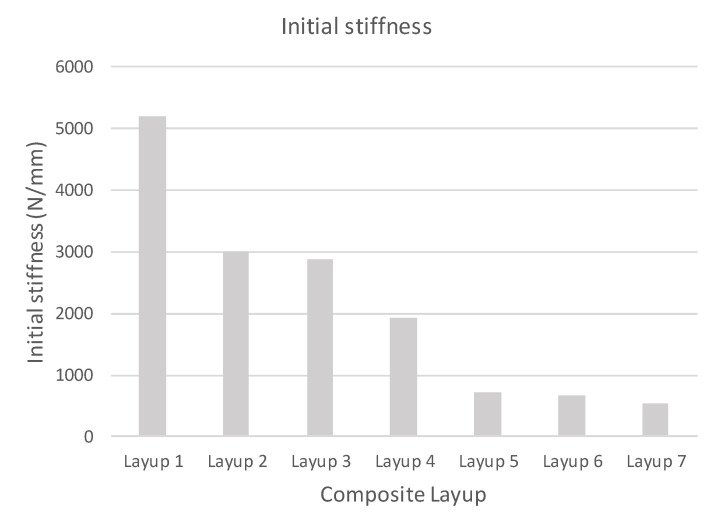
Initial calculated stiffness of specimen.

**Figure 11 polymers-14-01123-f011:**
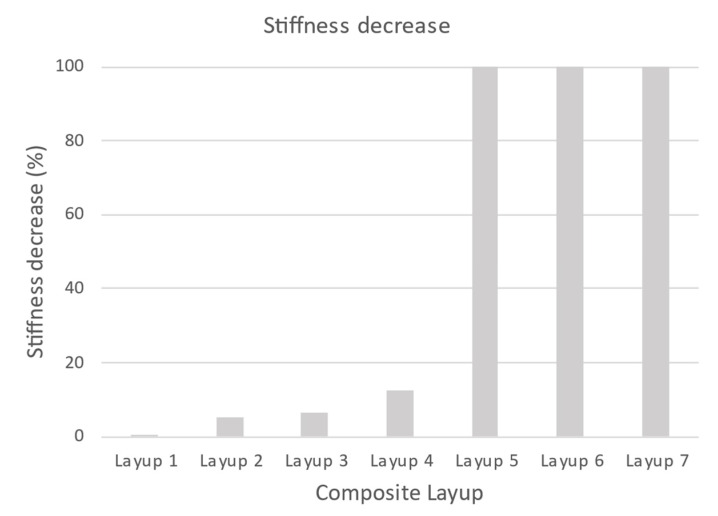
Decrease in stiffness depending on the composition of the composite.

**Figure 12 polymers-14-01123-f012:**
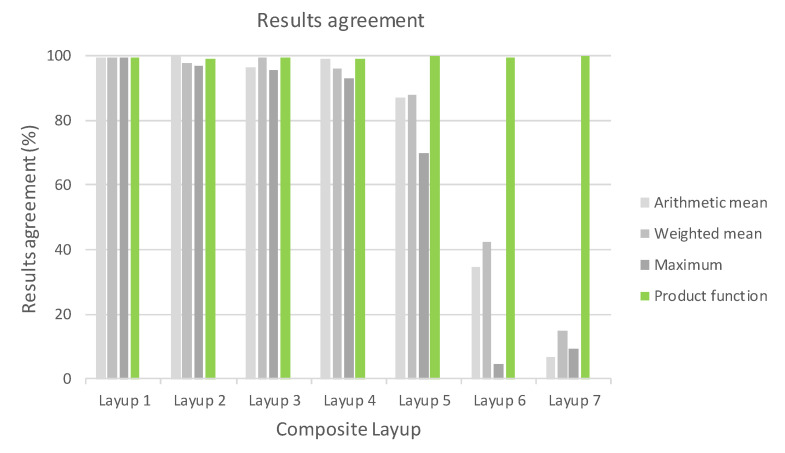
Consistency of results with reference calculation.

**Figure 13 polymers-14-01123-f013:**
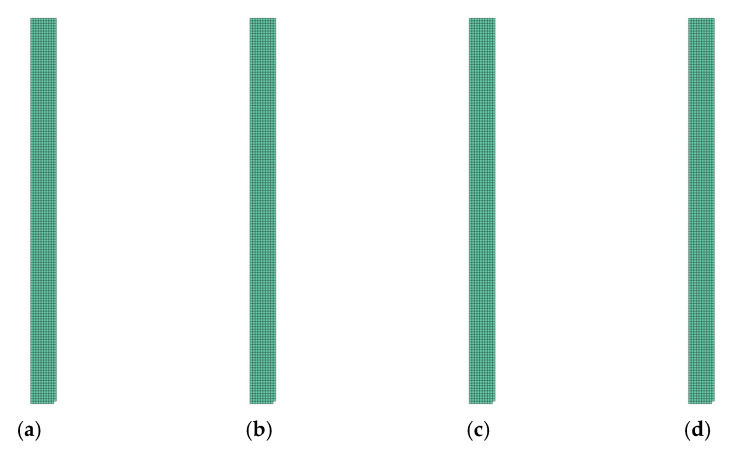
Damage of model Layup 4; Ply oriented at 0°; Selection ($d_{1}\to 0,d_{2}\to 0$): (**a**) Arithmetic mean; (**b**) weighted mean; (**c**) maximum; (**d**) product function.

**Figure 14 polymers-14-01123-f014:**
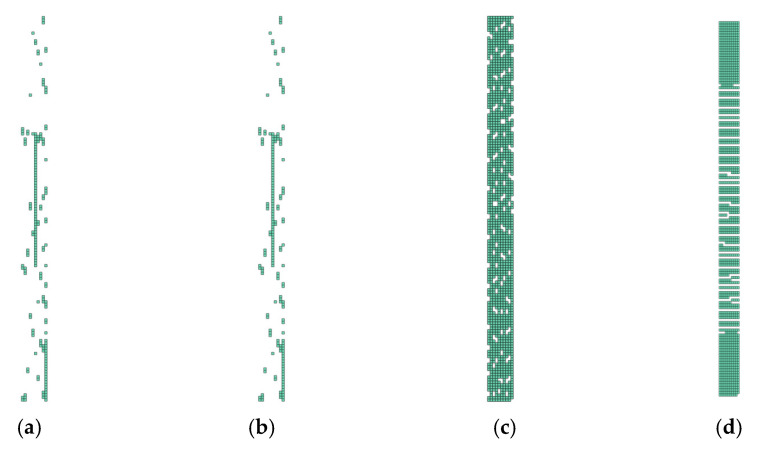
Damage of model Layup 4; Ply oriented at +45°; Selection ($d_{1}\to 0,d_{2}\to 0$): (**a**) Arithmetic mean; (**b**) weighted mean; (**c**) maximum; (**d**) product function.

**Figure 15 polymers-14-01123-f015:**
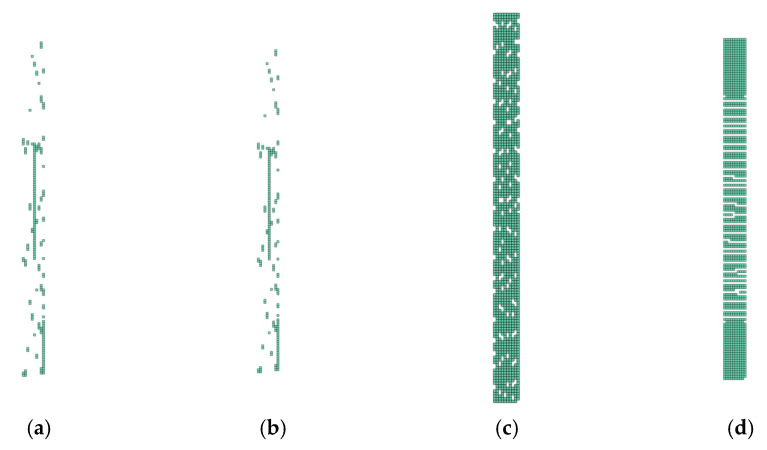
Damage of model Layup 4; Ply oriented at −45°; Selection ($d_{1}\to 0,d_{2}\to 0$): (**a**) Arithmetic mean; (**b**) weighted mean; (**c**) maximum; (**d**) product function.

**Figure 16 polymers-14-01123-f016:**
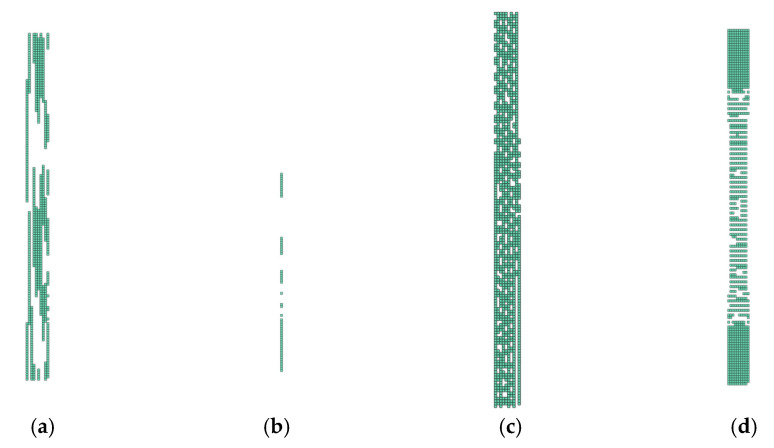
Damage of model Layup 4; Ply oriented at 90°; Selection ($d_{1}\to 0,d_{2}\to 0$): (**a**) Arithmetic mean; (**b**) weighted mean; (**c**) maximum; (**d**) product function.

**Table 1 polymers-14-01123-t001:** Damage parameter’s consolidation schema.

Function	Consolidate Parameter	Consolidate Variable
F1	$\left\{d_{1}^{+},\;d_{1}^{-}\right\}\;\to \;d_{1}$	$F1:\left\{E_{1}^{+}(d_{1}^{+}),\;E_{1}^{-}(d_{1}^{-})\right\}\;\to \;E_{1}^{d}\left(E_{1}^{0},d_{1}\right)$
F2	$\left\{d_{2}^{+},\;d_{2}^{-}\right\}\;\to \;d_{2}$	$F2:\left\{E_{2}^{+}(d_{2}^{+}),\;E_{2}^{-}(d_{2}^{-})\right\}\;\to \;E_{2}^{d}\left(E_{2}^{0},d_{2}\right)$
F12	$\left\{d_{1}^{+},\;d_{1}^{-},d_{2}^{+},\;d_{2}^{-}\right\}\;\to \;d_{12}$	$F12:\left\{G_{12}\left(d_{1}^{+},\;d_{1}^{-},d_{2}^{+},\;d_{2}^{-}\right)\right\}\;\to \;G_{12}^{d}\left(G_{12}^{0},d_{12}\right)$

**Table 2 polymers-14-01123-t002:** List of damage consolidation functions.

Method	Consolidate Condition	
Arithmetic mean	$d_{i}=d_{i,a}$	$i=\left\{1,2\right\}$
Weighted mean	$d_{i}=\left\{\begin{array}{cc} 0, & {\mathrm{if}\;\|d\|}_{i}=0\\ d_{i,w}, & {\mathrm{if}\;\|d\|}_{i}>1 \end{array}\right.$	
Maximum	$d_{i}=\mathrm{Max}\left\{d_{i}^{+},\;d_{i}^{-}\right\}$	
Product function	$d_{i}=1-\left(1-d_{i}^{+}\right)\left(1-d_{i}^{-}\right)$	
Shear function	$d_{\mathrm{ij}}=1-\left(1-d_{i}\right)\left(1-d_{j}\right)$	$i,j=\left\{1,2\right\};\;i\neq j$ $i,j=\left\{1,3\right\};\;i\neq j$ $i,j=\left\{2,3\right\};\;i\neq j$

**Table 3 polymers-14-01123-t003:** Stability criteria.

Plane Stress	Shear-Bending Coupling
$E_{1}^{d},E_{2}^{d},\;G_{12}^{d}>0$	$E_{1}^{d},E_{2}^{d},\;G_{12}^{d},\;G_{13}^{d},\;G_{23}^{d}>0$
ν12d<E1dE2d12	

**Table 4 polymers-14-01123-t004:** Elastic properties of damaged composite.

$E_{1}^{d}$	$E_{2}^{d}$	$\nu_{12}^{d}$	$\nu_{21}^{d}$	$G_{12}^{d}$
(MPa)	(MPa)	(1)	(1)	(MPa)
$E_{1}^{0}\left(1-d_{1}\right)$	$E_{2}^{0}\left(1-d_{2}\right)$	$\nu_{12}^{0}\left(1-d_{1}\right)$	$\nu_{21}^{0}\left(1-d_{2}\right)$	$G_{12}^{0}\left(1-d_{1}\right)\left(1-d_{2}\right)$

**Table 5 polymers-14-01123-t005:** Elastic properties of laminate.

$E_{1}^{0}$	$E_{2}^{0}$	$\nu_{12}^{0}$	$G_{12}^{0}$	$G_{13}^{0}$	$G_{23}^{0}$
(MPa)	(MPa)	(1)	(MPa)	(MPa)	(MPa)
129,840	13,340	0.26	4890	4890	4630

**Table 6 polymers-14-01123-t006:** Strength of laminate.

$F_{1,u}^{+}$	$F_{1,u}^{-}$	$F_{2,u}^{+}$	$F_{2,u}^{-}$	$F_{12,u}$	$F_{13,u}$ $\left(F_{23,u}\right)$
(MPa)	(MPa)	(MPa)	(MPa)	(MPa)	(MPa)
2965.41	2911.81	100.88	109.42	100.76	98.41

**Table 7 polymers-14-01123-t007:** Damage evolution parameters.

$U_{1}^{+}$	$U_{1}^{-}$	$U_{2}^{+}$	$U_{2}^{-}$
(mJ/mm^2^)	(mJ/mm^2^)	(mJ/mm^2^)	(mJ/mm^2^)
35.56	34.28	0.92	1.08

**Table 8 polymers-14-01123-t008:** List of tested laminates.

Layup Id.	Layup Design	Layup Id.	Layup Design
1	[0°/0°/0°/0°]	5	[45°/−45°/−45°/45°]
2	[0°/+45°/−45°/0°]	6	[45°/90°/90°/45°]
3	[0°/90°/90°/0°]	7	[90°/90°/90°/90°]
4	[0°/+45°/−45°/90°]		

**Table 9 polymers-14-01123-t009:** Residual stiffness deviance from the reference results.

Consolidate Function	Layup 1	Layup 2	Layup 3	Layup 4	Layup 5	Layup 6	Layup 7
[−]	[%]	[%]	[%]	[%]	[%]	[%]	[%]
Arithmetic mean	0.38	0.03	3.75	0.81	12.94	65.19	93.05
Weighted mean	0.38	2.17	0.58	4.09	12.03	57.44	85.23
Maximum	0.38	3.24	4.51	6.98	30.23	95.41	90.50
Product function	0.38	1.05	0.66	1.06	0.02	0.50	0.08

## Data Availability

The data presented in this study are available on request from the corresponding author.

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
