# Peer review of "Linearization of Composite Material Damage Model Results and Its Impact on the Subsequent Stress–Strain Analysis"

_polymers, 2022, doi:10.3390/polym14061123_

Round 1

Reviewer 1 Report

In the submitted manuscript, the authors presented novel approach for a linearization procedure for numerical simulation of failure in composite structures. In the introduction, the authors presented an overview on problems of implicit/explicit numerical analyses and properly defined their research problem. However, some extensions to the Introduction are recommended. In section 2, the authors presented a workflow scheme of their approach for linearization of nonlinear phenomena simulated using FEA. Next, in section 3, the basic formulation of modelling damage in composite structures was presented. Further, the authors presented their approach based on adaptation of Hashin failure criterion to the linearized solving method and the corresponding formulation of consolidation functions. In section 6, an approach for generation of material cards with less demanding computational efforts was introduced. Section “Material properties” presented material properties of a test problem and requires extensions. In section 7, the scenarios of damage variation and comparison of various functions used for simulation of damage variation were presented and discussed. Next, the test problem was described. This description also needs to be detailed. Finally, the results of simulation for various consolidation functions was presented. Although the manuscript is original and interesting, it requires deep revision according to the comments below.

1) It would be beneficial to discuss real case studies of failure of composite structures, where the proposed scheme might be applied. This will strengthen and justify the authors’ motivation.

2) Please explain the reasons and justify selection of functions presented in Table 2.

3) Section 4 is defined twice (line 188). Please correct and renumber further sections.

4) It is recommended to describe the test problem in details, i.e. which composite was simulated, how the structure was loaded, how the damage appeared and evolved in this structure, etc. This is also the case of section 8.

5) Please discuss advantages (like computational time) and disadvantages (accuracy) of the proposed solving scheme.

6) It is recommended to compare the developed linearized method with the nonlinear method and evaluate in the quantitative way the effectiveness of the proposed approach.

7) Please support conclusions with quantitative results obtained in the study.

8) Please double-check the manuscript for minor typos and grammar errors.

Author Response

Thank you for the review. We have responded to all your points in the attached file.

Reviewer 2 Report

Reviewed article concerns linearization of composite material damage model results and its impact on the subsequent stress-strain analysis and is write in accordance with generally accepted standards of the scientific works. After careful reading of the submitted text there are some substantive remarks that should be taken into consideration by the Authors to improve reviewed text.

  1. At the end of the introduction should be clearly and concise given aim and the novelty in presented investigation.
  2. The Authors should integrate sections 2-8 as research methodology.
  3. The Results section provides results on Figs. 10-13 but with only 6 sentences of its analysis. Deeper scientific consideration of obtained results referred to the basic phenomena in analyzed processes should be given.
  4. I suggest also to give wider description of potential use of presented findings in scientific research as well as in industrial practice.
  5. In the discussion section should be provide more references to already known results from literature.
  6. The strengths and limitations of the obtained results and applied methods should be clearly described.
  7. I suggest providing the main conclusions as numbered sentences and refer to specific values (results of analysis) as well as basic phenomena that cause described results.
  8. The conclusions should highlight the novelty and contribution to the state of the knowledge in given area.

Author Response

Thank you for the review, we have responded to all your points.

Reviewer 3 Report

This paper is on numerical analysis of fiber reinforced polymer composites. Damage model was used to analyze the tensile strength changes. Consolidated damage parameters were generated to correlate and evaluate the extent of the mechanical strength variation. Readers should have some fundamental interest  in the paper.  The experimental work compliment the model analysis which makes the paper comprehensive. Some minor changes are suggested.

  1. Check the language expression all over the paper.
  2. The last sentence in the Abstract may be re-written and move to the Introduction section.
  3. Check the damage parameter plots Figs. 4-7, make sure that the marks on both x- and y-axis are correct. Also check the units on both axis.
  4. In Fig.9 ,the step time unit on x-axis is not shown.
  5. Rearrange Table 5 is recommended by swap the row and column.
  6. Check the reference format to make sure matching the journal's requirements.

Author Response

Dear reviewer, thank you for the comments. You can see our response below:

1.Check the language expression all over the paper.

Re: Will be checked using MPDI language editing service.

2.The last sentence in the Abstract may be re-written and move to the Introduction section.

Re: This sentence was added due to reviewers 1 and 2.

3. Check the damage parameter plots Figs. 4-7, make sure that the marks on both x- and y-axis are correct. Also check the units on both axis.

Re: It was checked and changed. Modifications are highlighted in the text.

4. In Fig.9 ,the step time unit on x-axis is not shown.

Re: It was corrected and changed. Modifications are highlighted in the text.

5. Rearrange Table 5 is recommended by swap the row and column.

Re: The table was not changed, because we believe it saves the space of document.

6. Check the reference format to make sure matching the journal's requirements.
Re: There were found some inconsitencies such as size of the text. It was corrected.

Round 2

Reviewer 1 Report

The authors made answered on the comments provided in the first round of review and made appropriate changes and extensions in the manuscript. I recommend to consider the manuscript for a publication in its present form.

Author Response

Dear referee,

thank you for review of the revised article.

Kind regards,

Martin Kadlec

Reviewer 2 Report

The authors disregarded a number of my remarks, as a result of which the introduced changes did not contribute to improving the scientific value of the article.

Author Response

Dear referee,

we are sorry that you are not satisfied with the article. We responded to your last comments as we thought would be the best for the article comprehension.

Kind regards,

Martin Kadlec
